# Liquid Biopsy for Detection of Pancreaticobiliary Cancers by Functional Enrichment and Immunofluorescent Profiling of Circulating Tumor Cells and Their Clusters

**DOI:** 10.3390/cancers16071400

**Published:** 2024-04-02

**Authors:** Andrew Gaya, Nitesh Rohatgi, Sewanti Limaye, Aditya Shreenivas, Ramin Ajami, Dadasaheb Akolkar, Vineet Datta, Ajay Srinivasan, Darshana Patil

**Affiliations:** 1Department of Clinical Oncology, Cromwell Hospital, London SW5 0TU, UK; 2Department of Medical Oncology, Fortis Memorial Research Institute, Gurugram 122002, HR, India; 3Department of Medical and Precision Oncology, Sir HN Reliance Foundation Hospital and Research Centre, Mumbai 400004, MH, India; 4Department of Medicine, Medical College of Wisconsin, Milwaukee, WI 53226, USA; 5Department of Oncology, The Royal Free Hospital, London NW3 2QG, UK; 6Department of Research and Innovation, Datar Cancer Genetics, Nasik 422010, MH, India; dadasaheb.akolkar@datarpgx.com (D.A.); drvineetdatta@datarpgx.com (V.D.); ajays@datarpgx.org (A.S.); drdarshanap@datarpgx.org (D.P.)

**Keywords:** pancreaticobiliary, pancreas, gallbladder, bile duct, cancer, diagnostic triaging, diagnosis, detection, circulating tumor cells, non-invasive, liquid biopsy

## Abstract

**Simple Summary:**

Cancers of the pancreas, gallbladder and bile duct (pancreaticobiliary tract) are often aggressive with poor survival. However, these cancers are not easily identifiable or detected because of vague (non-specific) symptoms and the absence of accurate blood tests. The diagnosis of these cancers involves imaging or endoscopy followed by sampling (biopsy) of suspected tissue for histopathological examination (HPE). Where the tissue sample is not of acceptable quality or is insufficient, the diagnosis (and treatment) can be hindered or delayed. In this manuscript, we describe a blood test that can detect pancreaticobiliary tract cancers with high accuracy. The test may be able to facilitate timely diagnosis of these cancers in suspected patients, especially where tissue-sampling-based diagnosis has been inconclusive or is not possible.

**Abstract:**

Circulating tumor cells (CTCs) have historically been used for prognostication in oncology. We evaluate the performance of liquid biopsy CTC assay as a diagnostic tool in suspected pancreaticobiliary cancers (PBC). The assay utilizes functional enrichment of CTCs followed by immunofluorescent profiling of organ-specific markers. The performance of the assay was first evaluated in a multicentric case-control study of blood samples from 360 participants, including 188 PBC cases (pre-biopsy samples) and 172 healthy individuals. A subsequent prospective observational study included pre-biopsy blood samples from 88 individuals with suspicion of PBC and no prior diagnosis of cancer. CTCs were harvested using a unique functional enrichment method and used for immunofluorescent profiling for CA19.9, Maspin, EpCAM, CK, and CD45, blinded to the tissue histopathological diagnosis. TruBlood^®^ malignant or non-malignant predictions were compared with tissue diagnoses to establish sensitivity and specificity. The test had 95.9% overall sensitivity (95% CI: 86.0–99.5%) and 92.3% specificity (95% CI: 79.13% to 98.38%) to differentiate PBC (n = 49) from benign conditions (n = 39). The high accuracy of the CTC-based TruBlood test demonstrates its potential clinical application as a diagnostic tool to assist the effective detection of PBC when tissue sampling is unviable or inconclusive.

## 1. Introduction

Early-stage pancreaticobiliary cancers (PBC) have better survival outcomes than late-stage disease [1]. However, most PBCs are detected at advanced stages (Stage IV) with poor prognosis where 5-year survival rates are <5% [2,3,4]. Globally, there are no recommendations for PBC screening in non-high-risk individuals. Individuals at higher risk for pancreatic cancers may be advised endoscopic ultrasound (EUS) based or computed tomography (CT) surveillance [5]. The United States Preventive Services Task Force (USPSTF) guideline cites the unavailability of accurate tests to detect PBC [6]. There is thus a large unmet clinical need.

Further, early-stage PBCs are also generally asymptomatic, whilst advanced cases tend to have non-specific symptoms such as abdominal pain, anorexia, and weight loss. This clinical conundrum is amplified by diagnostic challenges in patients suspected of PBC. The initial diagnostic work-up for suspected cases may include evaluation of serum cancer antigen 19-9 (CA19-9) and carcinoembryonic antigen (CEA), which have lower accuracy and are inconclusive. Hence, suspected cases are referred for medical imaging such as EUS, CT, endoscopic retrograde cholangiopancreatography (ERCP) or magnetic resonance cholangiopancreatography (MRCP), which can provide information on the localization of the suspected disease, but do not conclusively establish a diagnosis. Hence (image-guided) tissue sampling is performed for histopathological examination (HPE), which is the current gold standard to establish a conclusive diagnosis of PBC (or non-malignant conditions) [7,8]. Diagnostic challenges to HPE include the inability to obtain (sufficient) tissue samples due to the inaccessible location of the tumor, as well as incomplete or inconclusive HPE findings due to non-diagnostic tissue, which may be seen in up to 11% of suspected cases [9,10,11,12,13]. Further, about 31% of cases with negative HPE findings have been reported as false negatives [9,10,11,12]. EUS-guided biopsy rather than fine needle aspiration cytology (FNAC) has improved the diagnostic yield but is invasive and not without significant risks such as bleeding, sepsis, pain and discomfort to the patient. These challenges delay diagnosis and time to treatment due to resource constraints.

A noninvasive test with high sensitivity and specificity for the detection of PBC is an unmet clinical need that could also have potential for PBC screening. Evaluation of circulating tumor cells (CTC) [14] can facilitate more effective detection of PBC. CTCs are ubiquitously detected in solid tumors and are highly specific for malignancy. They are undetectable in individuals with non-malignant conditions such as benign or inflammatory conditions (pancreatitis or IgG4 disease). This may represent a diagnostic dilemma on imaging or serum tumor marker profiling. While all CTCs arise from a malignant tumor, they are a heterogeneous population with several atypical/hybrid phenotypes being detected in blood [15]. Despite this heterogeneity, CTCs have been observed to retain the molecular characteristic features of the primary malignancy, and it is proposed that their evaluation can provide non-invasively diagnostically relevant information on the underlying malignancy [16].

We describe a noninvasive liquid biopsy test (TruBlood^®^) that enriches and detects PBC-associated circulating tumor cells (CTCs) and their clusters from peripheral blood samples. This test is based on our previously described functional enrichment process to isolate CTCs as single cells as well as in clusters [17], which can be profiled by immunocytochemistry (ICC) to obtain diagnostically relevant information on underlying malignancy [18]. In suspected PBC cases, CTCs enriched from blood samples are evaluated by ICC to determine the expression of cancer antigen 19.9 (CA19.9), mammary serine proteinase inhibitor (Maspin) along with epithelial cell adhesion molecule (EpCAM), cytokeratins (CK), and the common leucocyte antigen (CD45). Positive Expression of CK and EpCAM and negative CD45 status confirm the epithelial origin of the CTC, while positivity for CA19.9 and/or Maspin suggests pancreaticobiliary tract origin. In the present manuscript, we present the clinical performance characteristics of the TruBlood^®^ test.

## 2. Methods

### 2.1. Study Participants and Samples

Biological samples mentioned in this manuscript were obtained from participants in three prospective observational studies, all of which are registered at the Clinical Trials Registry—India (https://ctri.nic.in/Clinicaltrials/ (accessed on 7 March 2024)). The TRUEBLOOD study (Trial ID CTRI/2019/03/017918, ongoing) enrolls patients diagnosed with or suspected of cancers and patients with benign (non-malignant) conditions. The CTC-based Cancer Detection and Diagnosis Study (Trial ID CTRI/2022/02/040373, February 2022—ongoing) enrolls suspected cancer cases. The RESOLUTE study (Trial ID CTRI/2019/01/017219, January 2019—ongoing) enrolls healthy asymptomatic adults with no prior diagnosis of cancer. All studies were previously approved by the Ethics Committees (EC) of the sponsor (Datar Cancer Genetics, DCG) as well as of the participating institute(s). All studies are conducted in accordance with the Declaration of Helsinki.

Written informed consent was obtained from all study participants before the collection of peripheral blood samples by venous draw into EDTA vacutainers. Leftover blood samples from patients who had availed of the sponsor’s services were also used (after obtaining EC approval and informed consent in each case). Blood samples from suspected cancer cases were collected prior to the tissue sampling. All sample identities were masked with unique 10-digit alphanumeric codes, stored at 2–8 °C and processed at the CAP- and CLIA-accredited facilities of the study sponsor, which also adheres to various ISO quality standards. The reporting of observational studies in this manuscript complies with STROBE guidelines [19].

### 2.2. Markers, Antisera and Cell Lines

Details of the markers employed by the test, as well as the antisera and cell lines used for method development or as controls, are provided in Appendix A. The purity of all cell lines used in the study was confirmed by periodic Short Tandem Repeat (STR) profiling. All cell lines were also periodically tested and verified to be Mycoplasma-negative.

### 2.3. Enrichment of Circulating Tumor Cells from Peripheral Blood

Blood samples were processed to enrich CTCs from white blood cells (WBCs) as described previously [17]. Briefly, WBCs were isolated from blood samples following lysis of red blood cells (RBCs) and then treated with a proprietary CTC enrichment medium (CEM), which induces cell death in all non-malignant cells while the apoptosis reluctant malignant cells (CTCs) survive. After CEM treatment, the surviving cells are retrieved for further use. The process is also explained in Appendix A.

### 2.4. Immunocytochemistry Profiling of CTCs

The process of ICC profiling of CTCs was described previously [18] and is also provided in Appendix A. Briefly, the CTCs are seeded into wells of an imaging-compatible multi-well plate, fixed and sequentially immunostained with cocktails of fluorophore-conjugated antibodies (Ab) against the markers. Marker-positive cells are then visualized using an appropriate fluorescence imaging system. A schema showing the various steps of the process, including CTC detection and ICC profiling, is depicted in Figure 1. Test samples received a positive classification based on the detection of CK+, CA19.9+, CD45- cells and/or CK+, Maspin+, CD45- cells, with or without detection of CK+, EpCAM+, and CD45- cells. Samples with any other findings received a negative classification.

### 2.5. Method Development and Validation

Comprehensive details of Method Development and Validation are provided in Appendix A. These include the determination of marker expression (marker specificity) in various cell types (Appendix A), impact of innate factors (age, gender, primary organ and stage) on marker expression (Appendix A), (non-)detectability of CTCs in non-malignant conditions (Appendix A), stability of CTCs in patient samples (Appendix A), sensitivity metrics of the test (Appendix A), including linearity and limit of detection, specificity metrics of the test including limit of blank (Appendix A), interference (Appendix A) and inter-operator agreements (Appendix A).

### 2.6. Case-Control Clinical Study

The performance characteristics of the test to detect and differentiate PBC cases from asymptomatic individuals were ascertained and established in a case-control study with samples from 360 participants, which included 188 recently diagnosed, therapy-naïve cases of PBC and 172 asymptomatic adults, the latter having neither prior diagnosis of any cancer, no current suspicion of cancer, being generally asymptomatic, with normal findings on ultrasonography (USG) of the abdomen and normal serum CA19.9 levels. Samples of the asymptomatic cohort were randomized into Training and Test Sets in a 70%:30% ratio. The PBC cases were first segregated by stage and then assigned to training and test Sets in the same ratio. All blood samples were processed for CTC enrichment and detection of markers (CK, EpCAM, CA19.9, Maspin, and CD45) by operators unaware of the clinical status of the samples.

The marker expression status in the training set samples (n = 132 PBC, n = 120 asymptomatic) was correlated with their clinical status to ascertain marker positivity in cancer samples as well as the absence of such marker-positive cells in asymptomatic samples. Next, the marker expression status of the Test Set samples (n = 56 PBC, n = 52 asymptomatic) was used to predict the clinical status. The concordance of the prediction with the actual clinical status was used to determine the sensitivity (rate of true positives), specificity (rate of true negatives) and accuracy (combined rate of true positives and true negatives) of the case-control cohort test set.

Subsequently, all training set and test set samples (PBC and asymptomatic) were digitally pooled, and random 30% samples from PBC (stage-wise) and asymptomatic patients were selected. The identities of these samples were re-masked and provided for the prediction of clinical status and the determination of sensitivity and specificity.

The above pooling–resampling–remasking–reanalysis was repeated successively to obtain 20 iterations of the test set, including the original assignment. At each of these iterations, the sensitivity and specificity of the prediction were determined. The median sensitivity across the 20 iterations of the cross-validation process, as well as the overall specificity and accuracy, was reported along with the 95% confidence interval (CI) and the range.

### 2.7. Prospective Clinical Study

The performance characteristics of the test to diagnose PBC and differentiate it from benign PB conditions (PBB) were ascertained and established in a prospective clinical study with blood samples from 88 patients with no prior cancer diagnosis and who presented with symptoms and radiological findings suspected of PBC. All patients underwent routine tumor tissue sampling for standard histopathological diagnosis. Pre-biopsy blood was collected from all patients and processed for CTC enrichment and detection of marker-positive cells. Based on the marker expression status, the samples were classified as positive or negative. The clinical status (histopathological diagnosis: malignant v/s benign) of the samples was then unmasked to the sponsor to evaluate the concordance of the CTC-based prediction model with the clinical status (histopathological diagnosis) and determine the performance characteristics of the test.

## 3. Results

### 3.1. Method Development and Validation

Summarized findings of the Method Development and Validation studies are provided in the respective Appendix A along with indicated Appendix A.

### 3.2. Case-Control Clinical Study

The performance characteristics of the test were evaluated in a case-control study. The study inclusion criteria are provided in Appendix A, and the study cohort demographics are provided in Appendix A. The asymptomatic cohort was a South Asian population with <0.003% reported PBC incidence (in age > 40 years). Since the asymptomatic participants were also required to have normal CA19.9 and USG, the risk of an underlying malignancy was further lowered. Samples in the case-control study were split into training and test sets (70:30) and evaluated in a stringent, blinded, iterative manner, which eliminated any risk of overfitting. The observations in the Training Set are provided in Appendix A.

In the absence of any positive findings among the asymptomatic samples (n = 52) in the Test Set, the specificity of the test (cancer v/s healthy) was 100% (95% CI: 93.15–100.00%). The median overall sensitivity for the detection of PBC (n = 56) was 96.4% (95% CI: 91.6–100%), and the overall accuracy was 98.2% (95% CI: 93.47% to 99.77%). The number of samples (cancer-wise, stage-wise and overall) and median sensitivities (stage-wise and overall) (along with 95% CI and range across the 20 iterations) are provided in Table 1.

### 3.3. Prospective Clinical Study

The performance characteristics of the test to differentiate benign pancreaticobiliary conditions from malignant were established in a second clinical study populated with 88 patients (45 females, 43 males, median age 51 years, age range: 21–81 years) presenting with clinical symptoms and radiological findings suspected of PBC. None of the patients had a prior diagnosis of cancer. Among the 88 patients, 39 were eventually diagnosed with PBB and 49 with PBC (AD). The demographic details of the study cohort are provided in Appendix A. The observations in the study samples are provided in Appendix A. The specificity of the test (cancer v/s benign) was 92.3% (95% CI: 79.13–98.38%). PB-CTCs were detected in 47 of 49 cancer samples, yielding an overall clinical sensitivity of 95.9% (95% CI: 86.02–99.50%), and the overall accuracy was 94.32% (95%CI: 87.24–98.13%). The two false negative samples included a case of Stage 1 pancreatic cancer and a case of Stage 1 gallbladder cancer. The three false positive samples included a case of cholecystitis with cholelithiasis, a case of heterotopic pancreas and a case of cystic fibrosis of the pancreas. Cancer-wise, stage-wise, and overall sensitivities are provided in Table 2.

## 4. Discussion

We describe a liquid biopsy test for pancreaticobiliary cancer detection based on multiplexed fluorescence ICC profiling of CTCs functionally enriched from a blood sample. The concordance of the test findings with histopathological diagnosis was used to determine the Sensitivity (i.e., positive percent agreement, PPA in cancers) and Specificity (i.e., negative percent agreement, NPA in non-cancers) in two clinical studies. The test could detect adenocarcinomas (AD) of the pancreas, gallbladder, and bile duct with high sensitivity (including for early stages) and differentiate PBC cases from healthy individuals, as well as individuals with benign conditions with high specificity, with a lower risk of false positives.

The test is based on the detection of CTCs, which are ubiquitous in PBC and undetectable in individuals with benign conditions of the pancreas and biliary tract. Evaluation of CTCs can thus facilitate the detection of PBC with higher sensitivity and higher specificity due to the relatively low risk of false positive findings. While prior studies have shown the presence of CTCs in pancreatic cancers [20,21,22,23,24,25], the technology platforms used in these studies are known to have lower specificity. Although CTC heterogeneity has been reported in the literature, the high accuracy of the test in clinical studies suggests that this was not a (significantly) confounding factor for the intended purpose of the test and the study. These studies profile CTCs for disease prognostication and are limited in their ability to provide diagnostic profiling or screening.

The functional CTC enrichment and detection process of our test has distinct advantages over epitope capture, which has lower sensitivity to capture and detect CTCs with low expression of target biomarkers (EpCAM and CK), as has been demonstrated in several prior studies [26,27,28,29,30,31]. This functional enrichment method can effectively detect CTCs as well as their clusters with no loss of sensitivity associated with the age or gender of the patient or the primary or stage of disease. This advantage is reflected in prior clinical studies for this platform, where high CTC detection rates were consistently observed across all target cancer types and stages. The test also reported reliable performance in samples from asymptomatic individuals or those with benign conditions.

This CTC-based approach also has advantages over profiling of circulating tumor nucleic acids in blood samples, the latter having lower sensitivities, especially at the early stages, which hinder meaningful clinical applications [32].

The performance characteristics of the test have potential for the dual utility of the technology platform, i.e., screening as well as diagnostic guidance/triaging. When used for screening, the test may be able to detect PBCs at earlier stages where the cancers may be more amenable to curative intent treatments. Detection of cancers at earlier stages combined with prompt treatment permits less aggressive treatment, leading to a better quality of life for the patient and is associated with significantly reduced mortality [33]. Early diagnosis can also significantly reduce the cost of treatment; the treatment cost of early-diagnosed patients was 25% to 50% lower than that of patients with advanced cancer [34]. Despite the lower incidence of PBC cancers, they are aggressive with poor outcomes and have a significant economic impact on the patients; hence, the asymptomatic/higher-risk populations stand to benefit from screening and the associated potential for improved survival from early detection. The utility of this test to support diagnostic pathways is especially appreciable in suspected cases with comorbidities preventing tissue sampling, non-diagnostic tissue, or inconclusive histopathological findings, as well as in patients with negative findings but clinical suspicion of malignancy. The test can minimize the dependence on tumor tissue sampling in suspected cases.

The test is intended to complement rather than substitute the standard diagnostic process. A review of each modality’s relative strengths and limitations highlights how the modalities can complement each other and mitigate the risks and limitations of each. The evaluation of serum cancer antigens is a low-risk procedure and does not require specialized facilities, but these have low sensitivity and specificity. Radiological imaging can detect abnormal tissue morphology and permits localization of suspected malignancy but is not conclusive for diagnosis. Hence, there is a need for confirmatory diagnosis on tissue samples which is the current gold standard for initiation of clinical management protocols. However, there are significant risks and limitations associated with tissue sampling for diagnosis (as covered in the Introduction). In this regard, the high accuracy of the test for detecting malignant conditions is a potential advantage since the test neither requires a tumor tissue sample nor is it dependent on any foundational tumor-tissue-based test. Other compelling benefits of the test include its (a) safety, since the test is non-radiological and noninvasive, and blood draws for the test have low inherent procedural risks; (b) convenience, since the blood draw can be performed at any primary healthcare center and does not involve appointments or wait times, (c) cost-effectiveness, since it requires no specialized facilities and can be integrated easily within existing clinical pathways, (d) low risk, since an inability of the test to perform as expected (i.e., test failure) does not deny the individual of the standard of care procedures or treatments, and, (e) versatility, being equally equipped to support detection of all PBC globally despite the relatively higher incidence of pancreatic cancers in the western hemisphere and relatively higher incidence of gallbladder cancers in the eastern hemisphere. Thus, the test is amenable to integration into the SoC diagnostic pathway, as depicted in Figure 2.

The test is based on evaluating CA19.9, Maspin, CK, and EpCAM status on CTCs. These markers (CA19.9, Maspin, CK, EpCAM) are a part of routine histopathology evaluations globally, and there is no evidence to suggest that these markers (or any other histological aspects) or the detection of CTCs are prone to ethnicity-associated interference. While future studies are planned to include participants with diverse ethnicities, the available literature does not support ethnicity as a confounding factor.

While CA19.9 may be detectable in non-malignant (epithelial/endothelial) cells, such cells have limited survival in the blood due to anoikis. Any incidentally surviving non-malignant epithelial/endothelial cells are eliminated during the CTC enrichment process and do not interfere with the test [17]. This approach has been used in prior studies to detect CTCs in other malignancies [35,36,37]. Even though CA19.9 or Maspin may individually be expressed in different cancer tissues (or CTCs from different cancers), detection of CA19.9- and Maspin-positive CTCs by the test is clinically interpreted against the context of an underlying suspicion of PBC. Thus, the study findings support the intended use of the test for diagnostic triaging in suspected cases of PBC.

The test is currently not intended for the detection of non(-adeno)-carcinoma subtypes, such as neuroendocrine tumors, which account for about 5% of PBC. However, future iterations of the test are envisaged to include additional markers for the detection of these subtypes. Further, the test is currently not intended to distinguish between pancreas, gallbladder, and bile duct cancers. A limitation of our overall study is that it does not include a prospective study in the asymptomatic population to evaluate the clinical utility of the test for PBC screening. We propose such a study going forward.

The strengths of our overall study include stringent clinical validation studies with (a) sample blinding to eliminate bias, (b) an iterative case-control design to eliminate the risk of over-fitting, and (c) prospective assessment of performance in suspected PBC cases.

## 5. Conclusions

The study establishes the feasibility of a liquid biopsy for the detection of pancreaticobiliary cancers based on the enrichment and characterization of circulating tumor cells (CTC). Further prospective studies are planned to establish its clinical utility as a screening tool in high-risk individuals or for diagnostic guidance in suspected cases in the respective intended use (IU) populations. These studies would also report the positive and negative predictive values (PPV, NPV) in addition to the accuracy metrics.

## Figures and Tables

**Figure 1 cancers-16-01400-f001:**
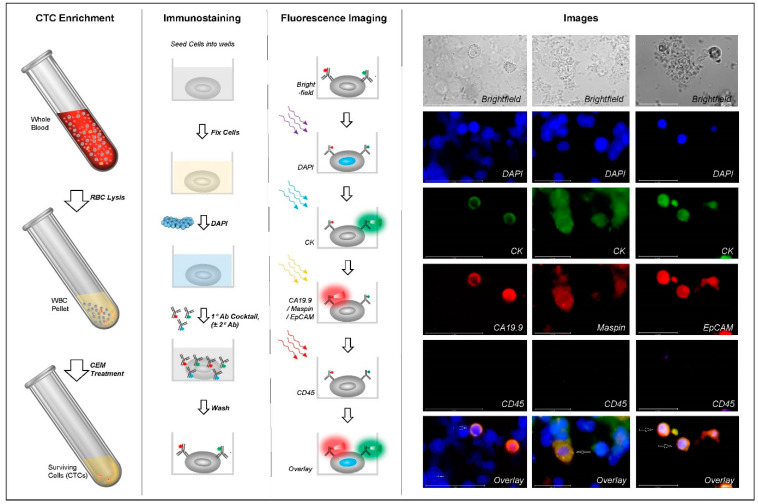
Schema of Test. Circulating tumor cells (CTCs) are then enriched from WBCs using a CTC enrichment medium (CEM) that eliminates all non-malignant cells and permits the survival of tumor-derived malignant cells (CTCs). The CTCs are seeded into imaging-compatible multi-well plates and immunostained using antibody (Ab) cocktails to detect the status of CA19.9, Maspin, CK, EpCAM, and CD45. Groups of wavy colored arrows under ‘Fluorescence Imaging’ represent fluorescence excitation while the gradient red and green regions represent fluorescence emission. Scale bars in the fluorescence images (right side) indicate 50 μm for CA19.9 and 25 μm for Maspin and EpCAM Note: Primary (1°) and secondary (2°) Ab steps are not shown separately.

**Figure 2 cancers-16-01400-f002:**
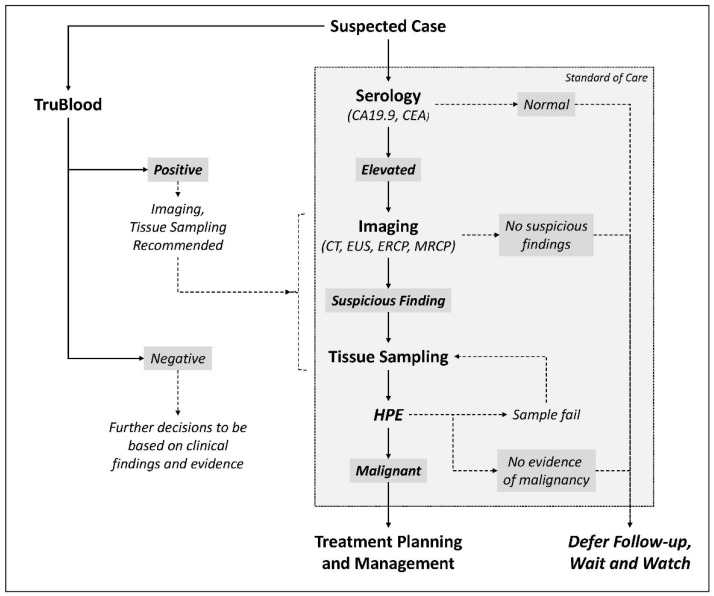
Trublood Testing in the Standard Diagnostic Pathway. The Trublood test described in the study is not intended to replace any of the tests in the standard diagnostic pathway but to complement the same and provide additional evidence that may be interpreted in conjunction with clinical findings and evidence from the standard investigations. The potential advantage of the TruBlood test is to facilitate clinical decision making among those individuals where tissue-based diagnosis is unavailable or unviable.

**Table 1 cancers-16-01400-t001:** Sensitivity of the Test in the Case-Control Study. The overall performance characteristics of the test were determined from 20 iterations of the test set. The table reports the median and range of stage-wise CTC detection rates and the number of (n) pancreas, gallbladder, and bile duct cancer samples, along with the cumulative (cancer-wise and stage-wise) Median Sensitivities. Values within parentheses adjacent to cumulative median sensitivities are the respective 95% Confidence Interval (95% CI) values.

	Pancreas	Gallbladder	Bile Duct	Cumulative (All Cancer Types)
Stage I	90.9%	100%	100%	86.7% (69.5–100%)
	Range: 63.6–100%	Range: 66.7–100%	Range: 100–100%	Range: 66.7–100%
	(n = 11)	(n = 3)	(n = 1)	(n = 15)
Stage II	100%	100%	100%	100% (100–100%)
	Range: 100–100%	Range: 100–100%	Range: 100–100%	Range: 100–100%
	(n = 12)	(n = 3)	(n = 3)	(n = 18)
Stage III	100%	100%	100%	100% (100–100%)
	Range: 100–100%	Range: 100–100%	Range: 100–100%	Range: 100–100%
	(n = 6)	(n = 3)	(n = 2)	(n = 11)
Stage IV	100%	100%	100%	100% (100–100%)
	Range: 100–100%	Range: 100–100%	Range: 100–100%	Range: 100–100%
	(n = 6)	(n = 4)	(n = 2)	(n = 12)
Cumulative (All stages)	97.1% (91.6–100%)	100% (100–100%)	100% 100–100%	96.4% (91.6–100%)
	Range: 88.6–100%	Range: 92.3–100%	Range: 87.5–100%	Range: 91.1–100%
	(n = 35)	(n = 13)	(n = 8)	(n = 56)

**Table 2 cancers-16-01400-t002:** Sensitivity of the Test in the Prospective Study. The table reports the CTC detection rates and stage-wise number of (n) pancreas, gallbladder, and bile duct cancer samples, along with the cumulative (cancer-wise and stage-wise) sensitivities. Values within parentheses adjacent to the cumulative sensitivities are the respective 95% Confidence Interval (95% CI) values.

	Pancreas	Gallbladder	Bile Duct	Cumulative (All Cancer Types)
Stage I	91.7%	88.9%	-	90.5% (69.6–98.8%)
	(n = 12)	(n = 9)		(n = 21)
Stage II	100%	100%	100%	100% (47.8–100%)
	(n = 2)	(n = 2)	(n = 1)	(n = 5)
Stage III	100%	100%	100%	100% (69.2–100%)
	(n = 4)	(n = 2)	(n = 4)	(n = 10)
Stage IV	100%	100%	100%	100% (75.3–100%)
	(n = 5)	(n = 6)	(n = 2)	(n = 13)
Cumulative (All stages)	95.7% (78.1–99.9%)	94.7% (73.9–99.9%)	100.0% (59.0–100%)	95.9% (86.0–99.5%)
	(n = 23)	(n = 19)	(n = 7)	(n = 49)

## Data Availability

The datasets used and/or analyzed during the current study are available from the corresponding author upon reasonable request.

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
