# Peer review of "Liquid Biopsy for Detection of Pancreaticobiliary Cancers by Functional Enrichment and Immunofluorescent Profiling of Circulating Tumor Cells and Their Clusters"

_cancers, 2024, doi:10.3390/cancers16071400_

Round 1

Reviewer 1 Report

Comments and Suggestions for Authors

The article discusses a new blood test, TruBlood, that can detect pancreaticobiliary cancers. This study found the test has high sensitivity and specificity, differentiating effectively between cancer and non-cancer cases. In general, this study is well-written. Here are the comments:

1.       The paper introduced a new method for detecting pancreaticobiliary cancers, but it lacks a direct comparison with existing standard tests, beyond a general statement. A more detailed comparison with existing diagnostic methods, including their advantages and limitations, would provide a clearer understanding of where TruBlood stands in the current diagnostic area.

2.       The number of iterations might depend on the variability of the results across these iterations. More iterations might be needed if there is significant variability. Please explain whether 20 iterations are sufficient to make a conclusion about the test's performance.

3.       While sensitivity, specificity, and accuracy are reported, other important metrics such as positive predictive value (PPV) and negative predictive value (NPV) are not mentioned. It would be beneficial to justify that those are not necessary to be included.

4.       The study included a South Asian population with a low reported incidence of PBC. The cohort might not be representative of other populations and could affect the generalizability of the results.

Author Response

We thank the Learned Reviewer for their comments and suggestions. Please see the attachment for the responses. 

Reviewer 2 Report

Comments and Suggestions for Authors

First of all, I would like to thank the authors for their work.

However, I do have some concerns regarding the manuscript:

The introduction of the article needs to be improved. It does not reflect the full issue. It has been shown that CTCs are a heterogeneous population of cells, except in addition, atypical/hybrid CTCs are found in the blood (doi: 10.1134/S0006297922040071).

The authors describe a liquid biopsy test for pancreatic cancer detection based on multiplex fluorescence ICC profiling of CTCs functionally enriched from a blood sample. However, they do not disclose the method of functional enrichment and ICC profiling of CTCs.

Section 2 (2. Methods), 2.4. Immunocytochemistry Profiling of CTCs needs to be clarified

For multiplex analysis, the authors used combinations of markers. It is not entirely clear what type of immunostaining was used: direct or indirect? If the direct type of immunostaining was used, the authors did not indicate which fluorescent labels were used in the case of monoclonal antibodies to (a) Anti-CK + Anti-CD45 + Anti-EpCAM, (b) Anti-CK + Anti-CD45 + Anti-CA19 .9, (c) Anti-CK, Anti-CD45, Anti-Maspin. If the indirect method of immunostaining was used, then it is also not entirely clear. It is necessary to indicate which primary monoclonal antibodies were used (clone, specificity, dilution, etc.). If secondary antibodies were used, then also provide information on them.

Supplementary Material S4 contains essentially the same information as in the article and does not reflect the method.

Based on schematic Figure 1, we can conclude that the direct method of immunostaining was used in the work, but labeled monoclonal antibodies were not indicated in the methods section. Moreover, in the additional materials, the authors write “Samples for CA19.9 and Maspin were incubated with secondary (2°) anti-mouse Ab.”, which contradicts Figure 1 and confuses the reader.

The units of measurement of the studied parameters are not indicated.

In Figures S1-S3 (Supplementary Figures), you need to add a note and Y-axis units. It is necessary to indicate how FI: fluorescence intensity was measured.

The authors conclude that there are pancreaticobiliary cancers (PBC) based on CTCs positive for the CA19.9 marker, but this marker does not have strict specificity. It is expressed on cancer cells of the stomach, liver, lung, colon and rectum. In addition, it can be expressed in normal tissue.

Authors are required to provide clinical and pathological characteristics of cancer patients, including histological diagnosis, TNM data, grade, ECOG status, etc.

Author Response

(The authors gave the same response as above.)

Round 2

Reviewer 2 Report

Comments and Suggestions for Authors

Хочу поблагодарить авторов за их работу. Авторы доработали и откорректировали рукопись с учетом замечаний и рекомендаций.